

# Insight from the association between critical thinking and English argumentative writing: catering to English learners' writing ability

Yanfang Hu[1,2] and Atif Saleem[3]

[1] College of Teacher Education, Zhejiang Normal University, Jinhua, Zhejiang, China
[2] Pinghu Normal College, Jiaxing University, Pinghu, Jiaxing, Zhejiang, China
[3] School of Education, Huaibei Normal University, Huaibei, Anhui, China

## ABSTRACT

**Introduction:** English argumentative writing (EAW) is a 'problem-solving' cognitive process, and its relationship with critical thinking has drawn attention in China. This is because fostering EAW proficiency is a crucial element but a challenging task for Chinese high school English teaching and learning. The present study examined how critical thinking is related to Chinese high school students' EAW performance. The study identified eight critical thinking disposition (CTD) subscales and aims to determine whether EAW and CTD are correlated.

**Methods:** A questionnaire modified from the Chinese Version Critical Thinking Disposition Inventory (CTDI-CV) and the Evaluation Criteria for English Argumentative Writing (ECEAW) were employed in this study. Both instruments were administered to 156 students from Grade 12. A purposive sampling of high school students was used in this study. Student EAW performance was scored by two experts based on the Evaluation Criteria for English Argumentative Writing.

**Results:** A significant relationship was found between students' CTD and EAW abilities. Furthermore, among the eight CTD subdispositions, cognitive maturity, truth-seeking, analyticity, and justice were found to be positively correlated with EAW, and they all were found to be the main predictors of EAW proficiency among high school students.

**Conclusion:** Zhangzhou high school students' CTDs were overall positive, and students' EAW performance correlated significantly with the overall CTD and its four subdispositions of cognitive maturity, truth-seeking, analyticity, and justice. These four subdispositions showed a significantly predictive validity on EAW performance as well.

Corresponding author
Atif Saleem, ad668@nenu.edu.cn

## INTRODUCTION

English writing is an intricate problem-solving process that requires not only basic writing skills but also the capacity to imagine, make claims, be visionary, and provide proper supporting subarguments (*Kirkland & Saunders, 1991*; *Bruning & Horn, 2010*;

*Howell et al., 2018*) for the claims, especially when the claims are argumentative. Additionally, as a critical and versatile skill (*Graham, 2006*), English writing is essential for academic success (*Al Asmari, 2013*) and required globally, in political and business contexts, serving as a benchmark for college admissions, job applications, and career promotions (*National Commission on Writing, 2004*). As an index of comprehensive English proficiency, English writing is also pivotal in countries such as China, South Korea, Japan, India, Netherlands, Sweden that study English as a second language. In China, an English writing task is a common assessment tool used on almost every standardized English test. However, English writing is a challenging and complex undertaking (*Anastasiou & Michail, 2013*), even for native speakers. Only one-quarter (24%) of students at both grades 8 and 12 in the United States perform at a proficient English writing level, and only 3% from both grades achieved an advanced level of writing proficiency, according to the American National Center for Education Statistics (*National Center for Education Statistics, 2012*). Chinese high school students also performed relatively poorly on English writing (*Liu, 2015*; *Bui & Luo, 2021*).

What factors could determine English writing proficiency? Some scholars indicated that writing process is a part of cognition and considered writing is best understood as a set of thinking processes, which writers orchestrate or organize during the act of composing (*Flower & Hayes, 1981*). The systematic cognitive research on the writing process started in the late 1970s, when *Flower & Hayes (1980)* applied cognitive psychological methods to investigate writing. They further emphasized that "think–and teach–writing" should be seen as "a problem-solving, cognitive process". Since the early 1990s, researchers have tried to discover the relations between English writing proficiency, language thinking, and writing style (*Kobayashi & Rinnert, 1992*; *Sasaki & Hirose, 1996*). It has become widely accepted that English passages are "linear", that means an English paragraph usually begins with a topic statement, and then proceeds to develop that central idea and relate that idea to all the other ideas in the whole essay by a series of subdivisions of that topic statement to prove something, or perhaps to argue something. While Eastern languages such as Chinese are "roundabout", taking a more indirect or circuitous approach to conveying information in which the development of a paragraph is "turning and turning in a widening gyre." The circles or gyres turn around the subject and show it from a variety of tangential views, but the subject is never looked at directly (*Kamimura, 1996*; *Yin, 1999*; *Wu, 2003*). Inspired by *Kamimura (1996)*, textual linguistics and discourse analysis methods have been widely adopted to study the effects of second language writing, including the structure (*Söter, 1988*; *Kirkpatrick, 1997*; *Kubota, 1998*), paragraph arrangement (*Ostler, 1987*; *Bickner & Peyasantiwong, 1988*), and characteristics of articulation and coherence (*Simpson, 2000*). The increasing use of computers directed people's attention to the factors of keyboard proficiency (*Barkaoui, 2016*), automatic scoring system (*Deane, 2013*; *Liao, 2016*) and different feedback types (*Hanjani, 2016*; *Latifi et al., 2021*). Currently, studies on self-efficacy (an individual's belief in his or her capacity to execute behaviors necessary to produce specific performance attainments) (*Sun et al., 2021*) and lexical bundles (recurring sequences of words or phrases that commonly occur together in a specific language or domain. These bundles are often

considered as fixed or semi-fixed expressions that have become established through frequent usage) (*Kim & Kessler, 2022*) are the main focuses.

Previous studies including *Devine, Railey & Boshoff (1993)*, *Deane et al. (2008)*, *Panahandeh & Asl (2014)*, *Decker et al. (2016)*, have proven that proficient performance in English writing involves various cognitive skills that in most cases are complicated for English learners in particular (*Peng et al., 2021*). The specificity of writing has been posited as the cause of this difficulty, since writing requires not only linguistic capability but also ideation and analytical capabilities, logic, and synthetic reasoning (*Anastasiou & Michail, 2013*; *Bruning et al., 2013*). Furthermore, some cognitive ability factors have been attributed to students' low writing performance.

As a part of cognitive competence, critical thinking skills have attracted researchers' attention. Studies have attempted to discover the effects of critical thinking skills on English writing (*Yang, Sun & Yin, 2016*; *Li, 2021*). However, the relationship has not been determined between the critical thinking disposition (CTD) and high school students' English argumentative writing (EAW) performance, and empirical studies in China are insufficient. Thus, to narrow this gap, the current researchers aimed to explore whether CTD is correlated with high school students' EAW performance. Hence, this study investigated eight CTD subscales as well as the relationship between EAW and CTD. The study aims to discovered CTD predictors of CTD for high school students' EAW abilities. Thus, three specific questions are addressed in this study:

To this end, this study sheds light on three research questions:

1) What are high school students' current CTDs in China?
2) Is there any significant relationship between high school students' CTDs and their EAW performance?
3) What are the predictors of the CTD on EAW performance?

## English argumentative writing (EAW)

Although writing in school includes a range of genres, the argumentative type is particularly significant (*Lin et al., 2020*). Improving and fostering argumentative writing performance is a vital component of English teaching reforms in schools and universities globally as well as a main challenge for teachers of English writing at the K–12 and college levels (*Newell et al., 2011*).

In the United States, EAW is emphasized as a passport to further educational and job opportunities (*Watt, 2010*). Similarly, in China, argumentation is one of the key assessment elements on English language proficiency, especially in the high-stakes college entrance examination, which plays an essential role in college admission decisions. Additionally, EAW tasks have been widely adopted in internationally renowned English general proficiency examinations, such as the International English Language Testing System (IELTS), the Test of English as a Foreign Language (TOEFL), and the Canadian Academic English Language Test (CAEL).

Regarding the IELTS, for instance, empirical findings have established that there is no correlation between argumentation writing and students' IELTS test scores (*Coffin, 2004*). However, the National Assessment of Educational Progress (NAEP) report revealed that approximately one-quarter of the students provide logical reasons in support of the examples they use in their argumentative essays, and students often fail to consider alternative perspectives (*National Center for Education Statistics, 2012*). These problems also occur in China (*Cai, 2017*; *Zhang, 2017*; *Cai, 2019*). As stipulated in the *Ministry of Education of the People's Republic of China (2018)*, Grade 12 students in China should be able to actively utilize resources to clearly express opinions in writing in a structured manner (*Ministry of Education of the People's Republic of China, 2018*). Thus, improving English argumentation proficiency is an important but difficult part of teaching and learning English writing in China.

From a cognitive perspective (*Hayes, 1996*; *Graham, 2018*; *MacArthur & Graham, 2016*), English argumentation is a process of problem-solving requiring self-regulation to reach the author's rhetorical targets (*Graham & Harris, 1989*). Writing proficiency is affected adversely by the inability to strategically allocate limited cognitive resources (*Ferretti & Fan, 2016*). Skilled writers write arguments based on their knowledge reserve of the topic, critical assessment standards, and argumentative discourse (*Ferretti & Lewis, 2019a*).

What are the critical evaluation standards on argumentative writing achievements? Previous studies including *Nimehchisalem & Mukundan (2011)*, *Paek & Kang (2017)*, *Ferretti & Graham (2019b)* and *Wang, Lee & Park (2022)* have inspired research about English argumentation and have promoted EAW performance assessments. An initial objective was to identify the linguistic features in high-quality writing samples (*Witte & Faigley, 1981*; *McNamara et al., 2015*), which Wen Qiufang and Liu Runzhou did. Based on close scrutiny of the 20 best compositions from 1–4 grades of English major undergraduates in China, the authors hypothesized four parameters (*i.e.*, relevance, explicitness, coherence, and sufficiency) accompanying the supposedly four thinking stages in writing (*i.e.*, topic comprehension, thesis statement development with supporting arguments, organizing a coherent discourse, and putting ideas into writing). Afterward, the authors tried to verify/falsify their hypotheses by marking another 100 compositions of the same kind twice over a 3-month period, and doing so yielded a framework for analyzing the general features of Chinese students' EAW and salient problems in the students' abstract thinking (*Wen & Liu, 2006*). The Evaluation Criteria for English Argumentative Writing (ECEAW) they constructed has been widely used in China (*Liu, 2013*; *Yang, 2014*; *Xu, 2016*; *Li, 2018*). However, until now, the analysis of English argumentation has mostly been at the undergraduate level, and little attention has been given to high school students in China.

## Critical thinking

An essential skill in education is critical thinking because it helps students to reflect on and grasp their own viewpoints. This skill allows students to use their own observations and expertise to make sense of things (*Raj et al., 2022*). Various definitions of critical thinking

have been given. For instance, *Glaser (1942)* defined critical thinking as "an attitude of being disposed to consider in a thoughtful way the problems and subjects that come within the range of one's experience; knowledge of the methods of logical inquiry and reasoning; and some skills in applying these methods". This definition considers critical thinking as a synthesis of attitude, knowledge, and skills. However, *Ennis (1987)* insisted critical thinking was "a reasonable, reflective thinking that is focused on deciding what to believe or do". He believed critical thinking contained both critical thinking silks and personality traits.

Critical thinking has also been described as "a mode of thinking, about any subjects, contents or problems, in which the thinker improves the quality of his or her thinking by skillfully taking charge of the structures inherent in thinking and imposing intellectual standards upon them" (*Paul & Binker, 1990*). These definitions reveal that critical thinking is a mode of thinking on the subjects within our realm of experience and helping us make decisions. Critical thinking should be reflective, reasonable, and logical, containing both critical thinking skills and personal dispositions. Peter Facione offered a more precise definition in the Delphi Report. It states that critical thinking is "a purposeful, self-regulatory judgment which results in interpretation, analysis, evaluation, and inference, as well as explanation of the evidential, conceptual, methodological, criteriological, or contextual considerations upon which that judgment is based" (*Facione, 1990*).

To this end, critical thinking consists of a learned collection of analytic thinking skills and a tendency to engage in the reasoning process (*Halpern, 2003*). Earlier studies have shown that the critical thinking disposition (CTD) is an inner motivation that guides decision-making and problem-solving, and that is essential for the application of critical thinking and the tendency to think critically (*Colucciello, 1997*; *Facione, Facione & Giancarlo, 2000*). Based on these studies, *Fesler-Birch (2005)* further found that CTD could be evaluated as a baseline for critical thinking performance.

The California Critical Thinking Disposition Inventory (CCTDI) (*Facione & Facione, 1992*) is one of the most well established instruments for assessing students' CTDs, designed on the definition of critical thinking formulated by Peter Facione in 1990. The CCTDI contains seven subdispositions with 75 items: inquisitiveness, self-confidence, truth-seeking, open-mindedness, analyticity, systematicity, and cognitive maturity (*Facione, Facione & Sanchez, 1994*). Since its development in 1990s, the CCTDI has been widely used in CTD studies (*Miri, David & Uri, 2007*; *Zuriguel-Pérez et al., 2017*; *Du et al., 2013*). The CCTDI has different versions. Luo and Yang were the first to translate the CCTDI into Chinese and to use it in in China (*Lou & Yang, 2001*).

After being revised twice, this version has good internal consistency and reliability. *Peng et al. (2004)* argued that, although the previous Chinese version of the CCTDI included semantic equivalence, it ignored cultural factors. Therefore, *Peng et al. (2004)* adapted and modified the CCTDI to obtain a conceptually equivalent Chinese variant that has the cultural sensitivity to be applied with Chinese-speaking students. However, *Peng et al. (2004)* chose nursing students to test the instrument's validity and reliability and doing so limits the questionnaire's generalizability. Therefore, *Wen et al. (2009)* retranslated the

CCTDI and constructed the CTDI–CV, mainly focused on checking the consistency of the Chinese translation with the original English, leaving no translation traces, and making the language consistent and smooth but not overly colloquial (*Wen et al., 2009*). The revised CTDI–CV contains 54 items with eight subdispositions: analyticity (the ability to independently and objectively analyze life problems and to foresee the outcome or consequences of an event based on facts), truth-seeking (the desire to seek the truth and to explore the essence of things), open-mindedness (tolerance and openness to external things and different perspectives), systematicity (the ability to overcome difficulties and solve problems with perseverance and an indomitable will), cognitive maturity (a measure of whether the understanding of things is comprehensive and life events are considered carefully), inquisitiveness (an instinct people have to be curious about the unknown), self-confidence (the trust in one's ability to do a certain thing well or solve a certain problem), and justice (conscious criteria for judging whether something is morally and legally fair) (*Wen et al., 2009*). Details have been shown in Table 1.

The CTDI–CV has generally been proven to be a reliable and valid instrument for assessing Chinese students' CTDs (*Wen et al., 2010a*, *2010b*, *2011*). However, the CTDI–CV has mostly been used for undergraduates or postgraduates and a preliminary analysis revealed that its internal reliability in this study was not acceptable. Therefore, the instrument was slightly modified for high school students to ensure its reliability and validity. Additionally, the reliability and validity analyses for the modified CTDI–CV were checked by educators and experts at the School of Educational Science in Minnan Normal University, China.

## EAW and CTD

English argumentation is a "problem-solving" cognitive procedure, demanding self-regulation to reach the author's targets (*Graham & Harris, 1989*), and critical thinking is "a purposeful, self-regulatory judgment which results in interpretation, analysis, evaluation, and inference, as well as explanation of the evidential, conceptual, methodological, criteriological, or contextual considerations upon which that judgment is based" (*Facione, 1990*). *Dong & Yue (2015)* have posited that English writing as a cognitive process is indivisible from the cultivation of critical thinking ability, and their study employing questionnaires and writing test has shown that students' critical thinking abilities are related to on their English writing performance. Therefore, improving the ability to think critically is essential for fostering English writing abilities (*Li, Gu & Qian, 2019*).

*Dong & Yue (2015)* showed that students' English writing proficiency is strongly influenced by their critical thinking skills, and suggested that cultivating students' critical thinking skill is necessary for improving their English writing competence. Since EAW depends on critical thinking ability to analyse facts, produce and organise ideas, maintain opinions, make comparisons, judge arguments, and solve problems by the use of existing information, previous knowledge, experience, and world knowledge when writing' (*Barnawi, 2011*). However, what is the relationship between them?

**Table 1  Definition and items of the Chinese version of California critical thinking disposition inventory (CCTDI-CV).**

| Disposition | Definition | Items | Unload and removed items |
|---|---|---|---|
| Analyticity | The ability to independently and objectively analyze life problems and to foresee the outcome or consequences of an event based on facts | 1. I will argue for my feelings or opinions about something, whether right or wrong.<br>2. If we disagree with others, we need to come up with reasons.<br>3. I really like to explore the nature of things.<br>4. I often can't help analyzing the process of other people's arguments.<br>5. I like to analyze complex problems methodically.<br>6. I prefer tests that require analytical thinking to memorized tests. | 1. Men and women have equal logical thinking skills. |
| Truth-seeking | The desire to seek the truth and to explore the essence of things | 1. I am not willing to choose between multiple, controversial points of view.<br>2. The so-called truth is nothing more than personal opinion.<br>3. When the majority agrees on something and the minority disagrees, I will support the majority.<br>4. Even if there is evidence that I am wrong, I will stick to my ideas.<br>5. It terrifies me to seek the truth about issues. | 1. In the case of most things, we can never understand their nature.<br>2. I only look for facts that support my opinion, regardless of facts that contradict my opinion. |
| Open-mindedness | Tolerance and openness to external things and different perspectives | 1. If a person's opinion is clearly wrong, he has no right to express his opinion.<br>2. I try to be less assertive, less judgmental about things.<br>3. I stand by my opinion and no one has the right to ask me for reasons.<br>4. There are many solutions to the problem, and I am not willing to analyze which is better.<br>5. Being open to different worldviews is less important than one might think.<br>6. People are entitled to their own opinions, but I don't have to listen to them. | 1. When I encounter a problem, I only seek advice from experts in my field, not from laymen. |
| Systematicity | The ability to overcome difficulties and solve problems with perseverance and an indomitable will | 1. Once I decide to do something, I don't give up easily.<br>2. When you encounter resistance to doing something, you think that maybe you're not cut out for it and give up.<br>3. My decisions are less susceptible to outside interference.<br>4. Many of my plans are difficult to achieve.<br>5. Achieving long-term goals is very difficult for me.<br>6. Faced difficulties, it is better to persevere than to find another way. | |
| Cognitive maturity | A measure of whether the understanding of things is comprehensive and life events are considered carefully | 1. The best basis for arguing for an opinion is how you feel at the time.<br>2. Being open to different opinions means not knowing right from wrong.<br>3. The best way to solve problems is to get answers from other people.<br>4. Life experience has taught me not to focus too much on logic.<br>5. My views on controversial topics are heavily influenced by the person I end up talking to. | 1. People think I'm too impulsive and hasty in making decisions.<br>2. Once the test results are not satisfactory, my enthusiasm for learning will be hit.<br>3. The essence of things is consistent with their appearance. |
| Inquisitiveness | An instinct people have to be curious about the unknown | 1. I am eager to learn challenging things.<br>2. Working hard to solve complex problems is a joy.<br>3. No matter what topic is discussed, I am eager to gain more understanding of it.<br>4. It is very important to me to understand other people's perspectives.<br>5. I try to learn as much as possible, even though I don't know when it will be useful.<br>6. Even at 60, I still want to learn new things. | |

(Continued)

Hu and Saleem (2023), *PeerJ*, DOI 10.7717/peerj.16435 

| Table 1 (continued) | | | |
|---|---|---|---|
| Disposition | Definition | Items | Unload and removed items |
| Self-confidence | The trust in one's ability to do a certain thing well or solve a certain problem | 1. As long as the test is prepared, I am not worried about failing.<br>2. I think I am capable of handling complex issues.<br>3. When facing a problem, my peers will come to me to make a decision because I can always make an objective analysis,<br>4. I can come up with creative solutions.<br>5. I am expected to set reasonable standards when making decisions.<br>6. When problems get tough, others expect me to take over. | 1. I appear to be logical, but I am not. |
| Justice | Conscious criteria for judging whether something is morally and legally fair | 1. If I witness a criminal robbery and called to testify in court, I will worry about getting in trouble.<br>2. There are too many things that violate laws and regulations, so we don't have to be angry about them.<br>3. I feel indignant when others are treated unfairly.<br>4. People always deal with problems based on their own interests.<br>5. I don't mind seeing other people cheating in exams.<br>6. When dealing with problems, we should try our best to be impartial, objective and unbiased. | |

**Note:**
SOURCE: *Facione & Facione (1992)* and *Wen et al. (2009)*.

Based on an analysis of 181 prospective teachers from six different departments in Turkey, *Bayat (2014)* found that the prospective teachers' critical thinking levels were related statistically significantly with their academic writing success. Similarly, a significant and positive relationship between college students' critical thinking skills and their English writing ability was found in China (*Wu, 2016*). Based on a study of 104 English major students, *Soodmand Afshar, Movassagh & Radi Arbabi (2017)* established a strong correlation between students' critical thinking skills and their English writing abilities. In addition, a significant relationship between the CTD and English writing has regularly been reported.

*McLean (2005)* claimed that a negative CTD accounts for a low writing proficiency. A study involving 73 senior English major students at a Shanghai university showed that the students were weak in CTDs and had comprehension difficulties as well as in demonstrating in-depth rhetorical clarity in academic English writing. This result implied a correlation between students' CTDs and their English academic writing performance (*Mu, 2016*). *Liu (2018)* explored 120 postgraduate students majoring in English and found a significant positive correlation between students' CTDs and their academic English writing. A positive linear correlation has also been found between critical thinking and English writing among secondary school students. *Jin (2021)* also examined 211 grade eight students' CTDs at the junior high school level and found that students demonstrated negative CTDs, which were positively correlated with their English writing achievements. Besides, *Liu (2021)* found a significant correlation between the CTDs of grade 12 students and their writing proficiency on English practical writing and continual writing tasks.

All the above discussions emphasize the importance of critical thinking to English writing, and some researchers further explored the relationship between critical thinking and English writing. In these studies, participants were mainly form college, and the types

of English writing involved were various, including picture writing, story writing, academic writing and so on. In summary, very few studies focus on the relationship between high school students' critical thinking and their performance on English argumentative writing. Hopefully, this study may bridge the gaps in the literature.

# MATERIALS AND METHODS

## Procedure and participants

This paper focus on high school student' critical thinking and their proficiency on English argumentative writing, so the population is all the high school students. A purposive sampling of high school students was used in this study. The reason for purposive sampling is the better matching of the sample to the aims and objectives of the research, thus improving the rigour of the study and trustworthiness of the data and results (*Campbell et al., 2020*). Because this study aimed to find out the relationship between critical thinking and English writing, it's better to take students with higher ability on critical thinking and English writing expression as participants, so that the association can be clearer and easier to be found. Since the development of critical thinking is limited by the level of cognitive development, critical thinking sprouts from childhood and get higher especially mature in senior grade of high school (*Ruggiero, 2012*). Considering this, a sample of 189 students from grade 12 students was involved in the study. All the participants were taken from a high school in Zhangzhou, China, because they were easily accessible to the investigators. Of the 189 questionnaires distributed to the students, 156 (84%) valid copies were returned.

Additionally, students were given 40 min to write a 120-word English argumentative essay on the same topic, "No smoking in public places?", which was prompted by sources from a relevant survey mentioned in the test (details in Table 2). In the writing, students were asked to show their opinions, defend sub-arguments and criticize counter-argument.

Two English teachers from Minnan Normal University scored the tests, and the average of the two scores was taken as the final score for students' EAW performance. The teachers had taught and studied English writing for over 13 years. A head teacher from Minnan Normal University who was specialized in English writing teachers' training and relevant researches was responsible for the evaluation and training. Before the formal scoring, the head teacher trained the two teachers based on ECEAW. After the training, the two teachers were asked to score some samples of EAW to test whether they have known the score criteria well. The result showed they have understood ECEAW well, and the scores given by them had no significant difference. Afterwards, the two teachers started to score the EAW from participants in two separate rooms to ensure the process was transparent. After scoring, the two teachers cross-checked all the scores, which the head teacher then rechecked and did not find significant difference. If there were, he would take careful. Then, since the authors have been studied English writing for several years and also specialize in English writing study, they cross-checked of everything to make sure the process and results were unbiased. This triangulation process ensured the reliability of the final scores. Table 3 presents the descriptive statistics of the final EAW scores.

**Table 2 English argumentative writing test.**

**EAW test**

A recent survey showed that 15% participants believed that people could smoke in public places, because they insisted that smoking was an individual freedom and could improve work efficiency. Additionally, smoking was a long-term habit that cannot be changed immediately. However, 85% participants supported banning smoking publicly, because they believed that smoking was unhealthy, money-consuming and also took its toll on the environment and other people. What's your opinion?

Please write a 120-word English argumentative essay on "No smoking in public places?" within 40 min. In this essay, please show choose one side and defend it. At the same time, the criticism of the other side also should be mentioned.

**Table 3 Students' English argumentative writing scores.**

|  | N | Minimum | Maximum | Mean | SD |
|---|---|---|---|---|---|
| Writing proficiency | 156 | 11 | 22 | 18.551 | 2.110 |

## Measures

A correlational research design was adopted to explore the relationship between the CTD (independent variable) and EAW (dependent variable). The CTD level was measured by the Chinese version of California Critical Thinking Disposition Inventory (CTDI-CV), and participants' performance on EAW was measured by the Evaluation Criteria for English Argumentative Writing (ECEAW).

## CTDI-CV

The CTDI–CV was adapted by *Wen et al. (2009)* from the CCTDI (*Facione & Facione, 1992*) and has been widely applied in the Chinese context (*Jin, 2021*; *Li, 2011*; *Ruan, 2012*; *Li, 2018*; *Lu, 2020*), mainly in studies involving English learners.

Harman's single-factor test had an explanatory variance for the first common factor of 25.76% is less than 40%, confirming no evidence of common method variance. Regarding the CTDI–CV questionnaire, eight subdispositions had 54 items, measuring the following subscales: analyticity (seven items), truth-seeking (seven items), open-mindedness (seven items), systematicity (six items), cognitive maturity (eight items), inquisitiveness (six items), self-confidence (seven items), and justice (six items). Each item was rated on a six-point scale of "strongly agree" (6), "agree" (5), "somewhat agree" (4), "somewhat disagree" (3), "disagree" (2) and "strongly disagree" (1), and the total scores of the CTDI–CV were between 54 and 324. Unloaded items were removed in the exploratory factor analysis (EFA) and the remaining items were retained for all eight factors. The items in systematicity, inquisitiveness, and justice remained the same. However, analyticity, open-mindedness, and self-confidence decreased to six items, and truth-seeking and cognitive maturity decreased to five items. Details are shown in Table 1.

The instrument reliability was analyzed using Cronbach's alpha, and the eight subscales in CTD showed reliability scores of 0.73, 0.71, 0.74, 0.70, 0.80, 0.72, 0.81, and 0.79. As all reliability scores were beyond the 0.7 threshold, the constructs were determined to be reliable (*Hancock & Mueller, 2013*; *Saleem et al., 2020*; *Byrne, 2016*).Construct validity

ensured the questionnaire's validity, and six factors were generated using an EFA. The results showed that the validity was acceptable (Kaiser–Meyer–Olkin test = 0.629 > 0.6; Bartlett's test of sphericity: $\chi^2$ = 2,665.49, df = 1,431, $p < 0.01$; factor loadings for all factor's items: 0.68–0.81; total variance: 66.39%, eigenvalue >1). Thus, the tool was reliable and valid.

### ECEAW

The ECEAW was determined by Wen and used to measure students' EAW proficiency and was divided into five levels (*i.e.*, best, good, moderate, poor, and bad) according to four parameters (*i.e.*, relevance, explicitness, coherence, and sufficiency) accompanying the supposed four thinking stages in English writing: topic comprehension, thesis statement development with supporting arguments, organizing coherent discourses, and putting ideas into writing (*Wen & Liu, 2006*). Table 4 provides the ECEAW details.

### Data analyses

The data analyses were performed using SPSS version 23 (SPSS Inc., Chicago, IL, USA). Chinese high school students' CTDs were approached using a descriptive statistical analysis, which illustrated the students' CTDs and eight subdispositions. Next, as this study focused on the relationship between the students' CTD and their performance on EAW, a Pearson correlation analysis was employed. It was followed to determine whether there was any significant correlation between the students' EAW proficiency and their CTDs as well as its eight dimensions. Last, in order to reduce interference between the variables, multiple regression analyses were conducted to examine the prediction of the students' CTDs subdispositions and their writing proficiency on English argumentation. The prediction of the different CTD dimensions for argumentative writing was explored in the regression analysis in detail.

During data screening, 33 questionnaires found to be incomplete and were thus removed. The instrument's face and content validity were ensured by educational experts from the School of Educational Science in Minnan Normal University, China. The data's internal reliability was determined by calculating Cronbach's α coefficients, and the construct validity was verified by conducting an EFA using the SPSS package.

### Ethical concerns and consent detail

Ethics committee approval was obtained from Zhejiang Normal University's institutional review board. The ethical principle of informed consent was upheld: each participant in the questionnaire was informed in advance of what was to be studied, and its possible benefits and impacts. All were informed of their right to withdraw their agreement to participate at any stage before the study was published. Finally, the researchers upheld the right to privacy by preserving the participants' anonymity at all points in the research process, ensuring that the publication of the research would not result in any conflicts of interest.

The two instruments involved in this study, namely The Chinese version of California Critical Thinking Disposition Inventory (CTDI-CV) and Evaluation Criteria for English Argumentative Writing (ECEAW) were used to measure participants' CTD level and their

**Table 4 Evaluation criteria for English argumentative writing (ECEAW).**

| Scoring range | Requirements |
| --- | --- |
| Fifth level (best): 21–25 point | Completes the test question task; covers all the main content points; the central thesis and subarguments are clear and appropriate; the subarguments are logically discussed, and the examples are appropriate and specific; the relationships among the subarguments are logical, clear, and definite. |
| Fourth level (good): 16–20 points | Completes the test question tasks; omits one or two subkey points but covers all the main content; the central thesis and most of the subtheses are clear and appropriate; individual subtheses are unclear; subtheses are logically discussed; examples are present but not specific; the discussion of subarguments is relatively logical, some of the arguments are specific, and some of the arguments have no examples; the relationships among the subarguments are logical and clear but not very definite. |
| Third level (moderate): 11–15 points | Basically completes the test questions; omits some content but covers all the main content; the central thesis is clear; some subtheses are relatively clear but some are unclear; the subtheses are clearly discussed, but the examples are too few or are not specific or appropriate; the relationship between the subthemes is clear but not logical enough. |
| Second level (poor): 6–10 points | Fails to complete the test questions properly; omits some of the main content, not described clearly, or irrelevant; the central thesis is relatively clear, but most of the subtheses are not clear or are not related to the central thesis; the subtheses are relatively clear but no examples or the examples are not specific or appropriate; the relationship between the subarguments is basically clear, but it takes the readers' effort to understand. |
| First level (bad): 0–5 points | Fails to complete the test questions, obviously omits the main content, and includes some irrelevant content that might be caused by misunderstanding the topic; the central argument and the subarguments are not clear; the reasoning is not definite, there are no examples, or the examples are inappropriate; the relationship between the subarguments is unclear or unconnected. |

performance on EAW. Both instruments are from Wen Qiufang, and the researchers have permission to use these instruments from the copyright holders/authors.

## RESULTS

Table 5 presents the descriptive statistics of the students' CTDs and the eight elements. Overall, the students' CTD was positive (M = 4.08 > 1.52). Among the eight dimensions, inquisitiveness (M = 4.41, SD = 0.51) scored the highest, while self-confidence (M = 3.62, SD = 0.46) scored the lowest. Besides, the students scored higher on justice (M = 4.38, SD = 0.52), cognitive maturity (M = 4.36, SD = 0.50), open-mindedness (M = 4.32, S = 0.44) and truth-seeking (M = 4.02, SD = 0.42) but lower on analyticity (M = 3.91, SD = 0.40) and systematicity (M = 3.63, SD = 0.43). The results also showed that five dimensions (inquisitiveness, justice, cognitive maturity, open-mindedness, and truth-seeking) had positive traits, while three dimensions (analyticity, systematicity, and self-confidence) had negative traits.

The Pearson Correlation analysis revealed that the CTD and EAW were significantly moderately correlated (r = 0.543, $p < 0.01$). In addition, EAW proficiency was significantly positively correlated with four CTD subscales: cognitive maturity (r = 0.529, $p < 0.01$), truth-seeking (r = 0.416, $p < 0.01$), analyticity (r = 0.348, $p < 0.01$), and justice (r = 0.185, $p < 0.05$). EAW proficiency was not significantly correlated at the $p = 0.05$ level with inquisitiveness (r = 0.333), systematicity (r = 0.856), self-confidence (r = 0.067), and open-mindedness (r = 0.888). The Pearson correlation also shows that there were some insignificant associations between CTD and EAW as it is depicted in Table 6.

**Table 5 Students' CTD dispositions.**

| Elements | Mean | SD |
|---|---|---|
| Analyticity | 3.91 | 0.40 |
| Inquisitiveness | 4.41 | 0.51 |
| Systematicity | 3.63 | 0.43 |
| Self-confidence | 3.62 | 0.46 |
| Truth-seeking | 4.02 | 0.42 |
| Cognitive maturity | 4.36 | 0.50 |
| Open-mindedness | 4.32 | 0.44 |
| Justice | 4.38 | 0.52 |
| Total | 4.08 | 1.52 |

**Table 6 Pearson correlation.**

| Constructs | AL | IQ | ST | SC | TS | CM | OM | JS | CTD | EAW |
|---|---|---|---|---|---|---|---|---|---|---|
| AL | – | | | | | | | | | |
| IQ | 0.076 | – | | | | | | | | |
| ST | 0.137 | 0.266** | – | | | | | | | |
| SC | 0.140 | 0.437** | 0.311** | – | | | | | | |
| TS | 0.127 | −0.068 | 0.028 | −0.015 | – | | | | | |
| CM | 0.131 | 0.194* | 0.010 | 0.158* | 0.324** | – | | | | |
| OM | 0.116 | −0.053 | 0.188* | −0.133 | −0.065 | 0.058 | – | | | |
| JS | 0.126 | 0.356** | 0.264** | 0.236** | 0.084 | 0.070 | 0.246** | – | | |
| CTD | 0.392** | 0.510** | 0.464** | 0.445** | 0.461** | 0.640** | 0.279** | 0.486** | – | |
| EAW | 0.348** | 0.078 | 0.015 | 0.147 | 0.416** | 0.529** | 0.011 | 0.185* | 0.543** | – |

Notes:
* $p < 0.05$.
** $p < 0.01$ (two-tailed).
AL, analyticity; IQ, inquisitiveness; ST, systematicity; SC, self-confidence; TS, truth-seeking; CM, cognitive maturity; OM, open-mindedness; JS, justice; CTD, critical thinking disposition; EAW, English argumentative writing.

In line with the prediction of the CTD on EAW performance, a multiple regression analysis is carried out to examine the extent to which the CTD can significantly predict EAW proficiency. As it is presented in Table 7, eight CTD subscales were the independent variables and EAW proficiency was the dependent one, while VIF results showde no evidence of collinearity. The R-square ($R^2$) of 0.436 and adjusted R-square ($R^2$) of 0.405 revealed four CTD subscales: cognitive maturity, truth-seeking, analyticity and justice accounted for 43.6% of the variance in EAW proficiency. The standardized regression coefficients (Beta) of 0.419, 0.257, 0.231 and 0.143 for cognitive maturity, analyticity, truth-seeking and justice, respectively, indicate that the four subscales significantly and positively predicted students' EAW performance ($p < 0.05$). This finding implies that high school students' EAW performance can be explained by the subdispositions of cognitive maturity, analyticity, truth-seeking and justice, among which cognitive maturity (Beta = 0.419) strongly predicts EAW proficiency. The analysis indicates the following

**Table 7 Regression analysis.**

R = 0.660
$R^2$ = 0.436
Adjusted $R^2$ = 0.405

| Model[a,b] | Unstandardized coefficients | | Standardized coefficients | | | |
|---|---|---|---|---|---|---|
| | B | Std. error | B (Beta) | T | p-value | VIF |
| (Constant) | 10.266 | 1.457 | | 7.048 | 0.000 | |
| Analyticity | 0.089 | 0.022 | 0.257 | 4.008*** | 0.000 | 1.071 |
| Truth-seeking | 0.079 | 0.023 | 0.231 | 3.391** | 0.001 | 1.205 |
| Cognitive maturity | 0.103 | 0.017 | 0.419 | 6.095*** | 0.000 | 1.231 |
| Justice | 0.073 | 0.036 | 0.143 | 2.015* | 0.046 | 1.319 |

Notes:
[a] Dependent variable: Writing proficiency.
[b] Predictors: (Constant) Analyticity, inquisitiveness, systematicity, self-confidence, truth-seeking, cognitive maturity, open-mindedness, justice.
* $p < 0.05$.
** $p < 0.01$.
*** $p < 0.0001$.

regression equation for the dependent and independent variables: "EAW proficiency = 10.266 + 0.419 * cognitive maturity + 0.257 * analyticity + 0.231 * truth-seeking + 0.143* justice".

## DISCUSSION

EAW performance is a major topic of interest in English teaching and learning, particularly in China's high schools. The present study explored the current CTD of Chinese high school students and the relationship between that and their EAW performance. The study also identified the CTD subdispositions that are positively related to and the main predictors of the high school students' EAW performance in China. Additionally, the study adds fresh evidence about the Chinese version of the CCTDI when applied in a non-Western context.

The results showed that the high school students' CTDs were overall positive (M = 4.08), that is in line with *Qing, Shen & Tian (2010)*, who examined the CTD of 121 grade 12 students in YuJin High School (M = 4.23), and *Li (2021)*, who found a positive disposition in grade 11 high school students (M = 4.095). These results revealed that high school students' CTDs have not improved dramatically during the past decade. However, after 3 years' further study in university, the students' CTD scores tended (M = 4.289) (*Liu, 2018*). This finding therefore contradicts (*Jin's 2021*) finding that junior school students' CTD at grade 8 is overall negative (M = 3.52). One reason is that the CTD is enhanced with age and learning, since the CTD is a psychological attribute that shapes one's beliefs or actions (*Profetto-McGrath et al., 2003*) enabling individuals to sufficiently solve problems and to make judgments as a product of thinking (*Facione & Facione, 2007*).

Compared with the CTD scores from other Asian, Africa and Middle Eastern countries—such as Israel (M = 4.02) (*Ben-Chaim, Ron & Zoller, 2000*), Turkey (M = 3.25 ± 0.27) (*Kaya, Şenyuva & Bodur, 2017*), Japan (M = 3.91) (*Kawashima & Petrini, 2004*) and Ghana (M = 3.95) (*Boso, van der Merwe & Gross, 2021*)—the result of this study is

relatively high (M = 4.08), and close to some developed countries such as Australia (M = 4.11) (*Tiwari, Avery & Lai, 2003*) and Italy (M = 4.10) (*Zoller et al., 2010*). This finding may partly challenge the statement that students from Asian societies (*vs.* those from non-Asian ones) are less inclined to demonstrate CTDs (*Wang et al., 2019*). However, room remains for improvement in comparison with other developed countries such as Norway (M = 4.72) (*Wangensteen et al., 2010*) and America (M = 4.33) (*Yeh & Chen, 2003*).

Additionally, the results also suggested that five dimensions (inquisitiveness, justice, cognitive maturity, open-mindedness, and truth-seeking) had positive traits, while three dimensions (analyticity, systematicity, and self-confidence) had negative traits. This showed that students had a strong interest in the unknown world, an inclusive attitude towards new knowledge, a relatively mature understanding about things and a passion for exploration, but they were not good at analyzing objectively and logically, lacking perseverance and confidence.

The current study reported a moderate relationship (r = 0.543, *p* < 0.01) between students' CTD and their EAW performance. These findings confirm those of earlier studies, such as *Li (2021)*, *Liu (2021)* and *Jin (2021)*. One reason is that the CTD correlates significantly with the total content knowledge resources and presentation strategies of English writing (*Yeh & Chen, 2003*). This finding indicates that students with stronger CTDs have wider content knowledge resources and presentation strategies, which are essential for good EAW performance. And among the eight subscales of CTD, cognitive maturity, truth-seeking, analyticity, and justice have positive correlation with EAW. This is because the four mentioned dispositions have direct influences on EAW, including the organization of writing, layout of sub-claims and examples, development of logical reasoning and so on. While the other four aspects, open-mindedness, systematicity, inquisitiveness and self-confidence have more invisible influence on critical thinking and indirect association with EAW. According to interviews, students who score highly on the CTDs perform better on the four thinking stages involved in EAW *i.e.*, topic comprehension, thesis statement development with supporting arguments, organization of a coherent discourse, and putting ideas into writing (*Liu, 2021*). For instance, understanding the task topic refers to the process of understanding concepts and judging the relationships among them. This process may involve the abilities of cognitive maturity and analyticity, since the former can help writers better understand the meaning of the title while the latter enables students to judge the relationships among concepts faster. Regarding developing a thesis statement with supporting arguments, which is central to writing, this process it is greatly influenced by the dispositions of truth-seeking and justice. The desire to seek the truth and explore the essence of things could drive students to carefully observe their surroundings, from which EAW's supporting arguments are usually derived. Moreover, the sense of justice could hone students' abilities draw distinctions, a skill that allow them to perceive or draw conclusions after thinking deeply about some social phenomena in daily life, and this process could be converted into a central EAW thesis statement. Meanwhile, the dispositions of systematicity, self-confidence, and open-mindedness have some effects on EAW that are not directly relevant, as they were

not significantly correlated at the 0.05 level. The disposition of inquisitiveness, which refers to 'an instinct that people are curious about the unknown' (*Wen et al., 2009*), help to expand students' knowledge reservoirs, but it does not help them to focus on exercising logical and critical thinking abilities. As a result, it had an insignificant relation with EAW performance.

The four related subscales (cognitive maturity, analyticity, truth-seeking and justice, respectively), were proved also have prediction on EAW proficiency. The other four subscales—inquisitiveness, systematicity, self-confidence, and open-mindedness—were not predictors, because they are not significantly related to EAW. The reason cognitive maturity, truth-seeking, analyticity and justice are significantly correlated and positively predictive of EAW was discussed in the context of the definitions of these four subdispositions and the EAW writing process.

Cognitive maturity refers to 'a measure of whether the understanding of things is comprehensive and life events are considered carefully', and truth-seeking is defined as 'the desire to seek the truth and to explore the essence of things' (*Wen et al., 2009*). Persuasive English argumentation requires an individual to 'find the essence of the topic' and to relate convincing subarguments and examples gleaned from the 'comprehensive and thoughtful understanding of things in life'. On the other hand, analyticity is defined as 'the ability to independently and objectively analyze life problems and to foresee the outcome or consequences of an event based on facts' (*Wen et al., 2009*), which is required throughout the argumentative writing process, specifically during the layout process. Justice is defined as 'conscious criteria for judging whether something is morally and legally fair' (*Wen et al., 2009*), and do help provide arguments in EAW writing, since the sense of justice can promote students to observe things around them objectively. These relations also can be found in the comparison between good and poor articles. For example, a student with high scores in these four dispositions gave three sub-argument to support his opinion "people shouldn't smoke in public places", from "Smoking is harmful to personal health and wealth" to "Smoking in public places violates the rights of others" and "Smoking in public places poses a significant fire hazard and thread public safety". From individual to others and to public group, the argumentation of the points of view was progressive. Meanwhile, the student used research data, news reports and celebrity quotes to support the sub-arguments. The whole structure of his EAW was logical and smooth. Additionally, during the argument, the student criticized the counter-arguments mentioned in the supplied material to strengthen the credibility of his opinion, such as "Although smoking could be seen as an individual right, public interest should be the most important thing in public places". While a student with low scores in these four dispositions even though also chose to defend "people shouldn't smoke in public places", but he only mentioned the sub-arguments from the resources in the test, from "Smoking is a pollution" to "Smoking is wasting money" and to "Smoking is harmful to the health". The logical correlations between these sub-arguments were not clearly articulated in the essays and some empty words were used to support the points which made the essay unconvincing.

Besides, a prominent feature of writing from the cognitive perspective is problem-solving (*Graham & Harris, 1997*), which is regarded as crucially important and

thought to positively affect EAW performance. Thus, a student with high CTD scores is expected to better gain the essence of the argumentative topic and comprehensively analyze the topic in a piece of EAW. According to this, we argue that cognitive maturity, truth-seeking, and analyticity, as the CTD components, could be strong EAW predictors. Therefore, it is helpful to enhance these CTDs to develop better EAW performance, since these were found to be linked to success in English argumentation.

## Limitations and future research

This study is limited in the research region and critical thinking aspects. First, the present study is limited to a developing, non-Western, Asian high schools. Considering this, high school students from other cities or relevant teachers should be involved in future study to deeply understand the relationship between CTD and EAW. Second, the current study is limited to the CTD, and other critical thinking aspects such as critical thinking skills have not yet to be explored. Incorporating other critical thinking factors in future studies could generate insightful results. Besides, the possible differences caused by years of study or other demographic factors need to be examined in future research.

## Conclusion and implications

EAW teaching and learning has been of prime importance for English education in China, since EAW performance is currently significant on both international and domestic English language proficiency tests. To discover the predictive influencing factors on EAW proficiency improve EAW performance, this study explored the relationship between the CTD (independent variable) and the EAW (dependent variable) proficiency of high school students with an emphasis on the CTD subscales. High school students' CTDs were overall positive, and students' EAW performance correlated significantly with the overall CTD and its four sub-dispositions of cognitive maturity, truth-seeking, analyticity, and justice. Furthermore, among the eight CTD subscales, only four dispositions (cognitive maturity, truth-seeking, analyticity and justice) showed a significantly predictive validity on EAW performance. The findings of the current study will contribute to the knowledge of Chinese high school students' cognition and English learning status. In addition, it has implications for the enhancement of EAW teaching and learning in China.

The findings showed that high school students in Zhangzhou, China generally have positive CTDs, *i.e.*, they perform well on the abilities of analyticity, truth-seeking, systematicity, open-mindedness, cognitive-maturity, inquisitiveness, self-confidence, and justice. In addition, their CTDs have been proven to be related to their performance on EAW. Specifically, their dispositions on cognitive maturity, truth-seeking, analyticity, and justice are related to their EAW proficiency score. A further analysis revealed that Chinese high school students' EAW performance can be predicted by their abilities in terms of cognitive maturity, truth-seeking, analyticity and justice. These results provide references for English teachers to improve students' English argumentative writing performance.

Primarily, in line with previous study findings in China (*Sun, 2020*; *Ren, 2020*), instructors in China should be concerned about students' CTDs, since students from China and other, more developed countries continue to have a gap. Secondarily, a

significant and positive correlation was found between EAW and CTD as well as its subdispositions—such as cognitive maturity, truth-seeking, analyticity, and justice—which has been confirmed in previous studies (*Han, 2020*; *Feng, 2021*). Therefore, instructors should provide clear CTD definitions for students and strengthen their critical thinking awareness. Lastly, teachers are urged to conduct suitable CTD training, especially on the four predictive subdispositions (*i.e.*, cognitive maturity, truth-seeking, analyticity and justice), which could foster and facilitate four thinking stages involved in EAW and directly improve high school students' EAW performance.

EAW is included as a prompt in the writing sections of some international standardized English exams (*e.g.*, TOEFL and IELTS) and English for Specific Purposes exams, which necessitate argumentative writing. Besides, EAW is a crucial skill in China because the performance on English argumentation regards as a key assessment element on English language proficiency, especially in the high-stakes college entrance examination, which plays an essential role in college admission decisions. Teachers of English writing in high school should focus on students' critical thinking and help them do a better job of analyzing the topic, establishing a layout, and organizing and writing argumentation logically, especially because EAW skills increasingly play crucial roles in students' general academics at all of their study levels (*Németh & Kormos, 2001*).

### Funding
The authors received no funding for this work.

### Competing Interests
The authors declare that they have no competing interests.

### Author Contributions
- Yanfang Hu conceived and designed the experiments, performed the experiments, analyzed the data, prepared figures and/or tables, and approved the final draft.
- Atif Saleem conceived and designed the experiments, analyzed the data, authored or reviewed drafts of the article, and approved the final draft.

### Ethics
The following information was supplied relating to ethical approvals (*i.e.*, approving body and any reference numbers):

College of Teacher Education, Zhejiang Normal University

### Data Availability
The raw data is available in the Supplemental File.

## Supplemental Information

Supplemental information for this article can be found online at http://dx.doi.org/10.7717/peerj.16435#supplemental-information.

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
