# Peer review of "Insight from the association between critical thinking and English argumentative writing: catering to English learners' writing ability"

_PeerJ, doi:10.7717/peerj.16435_

## Round 0.1 · original submission · Major Revisions

All three reviewers, while positive in general, raised issues that you should carefully and rigorously address. This will require considerable re-writing and further analysis. Their comments are explicit and must be addressed fully and carefully, with your cover letter explaining how you have modified your manuscript in accordance with their suggestions. Two have provided files for you to consider.

·

Basic reporting

Please see attached review

Experimental design

Please see attached review

Validity of the findings

Please see attached review

Additional comments

Please see attached review

·

Basic reporting

There are some minor grammatical issues that could be addressed (see below).
My opinion is that a lot of terminology used in the introduction section will need some additional explanation (for example, I'm not sure the readership will understand "self-efficacy ... and lexical bundles")
Examples of corrections could be:
p1: lines 21-23 - line 23 indicates plural but the previous sentences mention only 1 questionnaire: clarification required.
p1: the conclusions don't flow from the results - what does 'engaged in the context' mean?
p2: line 35 - is this English writing or EAW?
p2: line 42 - "English writing task", maybe - "an English..."?
p2: lines 56-58: explain what is meant by linear and roundabout text structure.
p3: line 81: "(the) argumentative type"
p3; line 93 - what test scores?
p4: lines 177 - 180: clarify this text as it indicates the CTDI-CV is both reliable and 'not acceptable'.
p7: lines 270-271 - check the formatting
p8: line 293: check the grammar.
p11: lines 434 - 439: check the grammar.

Experimental design

There are no issues I could identify with the design.

Validity of the findings

Two issues need to be clarified/resolved:
For table 4: why are the standard deviations larger than the means? Does this suggest that the data is non-parametric and so mean & sd should not be used?
p11: lines 450-451 (and subsequent paragraph) - why does a correlation/prediction indicate cause? It's not obvious what the "clear messages" are for improving EAW in school beyond that EAW and CTD are correlated. Is this evidence that improving critical thinking will improve EAW?

Additional comments

I'm not sure I agree with the subject sections (cognitive disorders and psychiatry)
The authors should clarify what is meant by the phrase "No public the data excluding raw data set" in the IRD jpg means, and confirm that in including the raw data table in the PeerJ supplemental data conforms to this requirement.
There are some issues that need resolving with the references:
A google search on the title of the Pan and Wu reference (2019) indicates that there are no hits. A google search on the title of the Lin reference (2016) indicates that there are no hits. A google search on the title of the Li reference (2015) indicates that there are no hits. Can this be explained? I stopped checking after I searched for these 3 references. Later I did find other references OK. Can a list be generated of all the references and their availability?

·

Basic reporting

Primarily, I have to disclose that almost all of my detailed comments are shown in the manuscript (using tack change). In line with report writing, the manuscript is mostly written in a clear and unambiguous manner. However, it needs revision for minor language usage errors. Besides, the introductory and background section of the manuscript also tries to show the research context. The literature is also well-referenced and relevant. Nonetheless, it would be more comprehensive if relevant studies from Africa are incorporated.
The structure of the manuscript also conforms to PeerJ standards and discipline norms. In addition, the supplementary files like tables are also relevant to the study. However, the data presented in Table 1 is not relevant. Hence, as I stated in the manuscript through the “track change”, the participants’ background information is collected for this study but not linked with the major research variable, critical thinking. Why is it? Thus, years of study and other demographic factors should be examined and reported in this study. If not, the demographic data should be removed from the paper because it is incorporated into the study for nothing.

Experimental design

As far as my knowledge is concerned, this article manuscript is original primary research that is within the scope of the journal. Besides, the research questions are well articulated, linked to the research title, and relevant to fill the knowledge gap related to critical thinking and argumentative writing. The methods section of the manuscript described the methodology aspects of the study with sufficient details. The study used relevant data-gathering instruments and data analysis methods with appropriate ethical standards. However, as I stated in the manuscript with “track change”, minor revision is needed to the methods section to make it clearer and reader-friendly. For instance, it would be better if the samples, sampling technique, and population are explained in the methods section.

Validity of the findings

The findings of the paper are valid because they are interlinked with the research questions, and they showed the results of the collected data. Besides, the necessary procedure is also followed in the study which indicates that the findings are valid. The conclusions of the study are also stated in a good manner and linked to the research questions.

Additional comments

By and large, the entire part of the paper needs revision. As I stated in the manuscript (using track change), the manuscript should be revised to clarify the theory of the study, complete citations, elaborate methods, fix reporting errors, and enhance the language usage including grammar, diction, choppy sentences, vague ideas, etc.

---

## Round 0.2 · Minor Revisions

Out of the 3 specific papers that the reviewer could not access, we still could not access one (Li, 2015). It appears that these papers will not appears in a Google search as they are written in the Chinese language, and so it becomes difficult/impossible for English-speaking readers to access those sources without the links the authors have provided in their table.

This needs to be addressed by the authors, ideally by providing an alternative English language citation.

That issue apart, thank you for your careful attention to the revisions. Once this issue is resolved I will accept the paper.

·

Basic reporting

I have conducted a check on the authors replies to the comments in my original review (see additional comments below).

Experimental design

I have conducted a check on the authors replies to the comments in my original review (see additional comments below).

Validity of the findings

I have conducted a check on the authors replies to the comments in my original review (see additional comments below).

Additional comments

The authors have amended the text and addressed the issues I raised in a clear an concise way. I appreciate their efforts.
I am unclear as to the issue over 'raw data and public release, but the authors comment that that statement allows them to release that data they have a match PeerJ's requirements

---

## Round 0.3 · accepted · Accept

Thank you for attending to the remaining issue. Indeed it is, as you observe, a pity we cannot access English translations of the original citations. However, thank you for addressing this point and I am now happy to recommend that we accept this work.